# Does Climate Change Influence Guest Loyalty at Alpine Winter Destinations?

**Thomas Bausch** [1] **, Andreas Humpe** [2] **and Stefan Gössling** [3,4,5,]*

1   Competence Centre Tourism and Mobility, Free University of Bozen-Bolzano, 39031 Bruneck-Brunico, Italy
2   Department of Tourism, Munich University of Applied Sciences, 80636 Munich, Germany
3   School of Business and Economics, Linnaeus University, 391 82 Kalmar, Sweden
4   Department of Service Management and Service Studies, Lund University, Box 882, 251 08 Helsingborg, Sweden
5   Western Norway Research Institute, P.O. Box 163, 6851 Sogndal, Norway
*   Correspondence: stefan.gossling@ism; Tel.: +39-0474-013609

**Abstract:** Research has dealt extensively with different aspects of climate change and winter tourism such as the impact on ski resorts and ski lift operators, adaptation strategies, governance at destinations and reactions of winter sports guests to changing snow conditions. This paper goes deeper into the question of destination choice and examines the role of climate change among the many factors affecting guest loyalty at Alpine winter destinations. The study uses an established destination choice model with choice sets, destination image and dynamic feedback loop. A qualitative online forum identifies factors influencing winter destination choice, followed by a quantitative survey which compares Alpine winter holidaymakers categorised as "loyal", "disloyal" and "undecided". The results demonstrate that climate change clearly influences destination choice, but snow sports are not the only affected attractors. Enjoyment of the natural environment and value for money are just as high on the list of guest motivators. This indicates that climate change adaptation measures such as snowmaking can be counterproductive to guest loyalty because they spoil the natural scenery and raise prices. The paper concludes with a recommendation for winter destinations to prioritize conservation of the natural environment and integrate more environmental protection measures into their management strategies.

**Keywords:** climate change; destination choice; destination loyalty; winter tourism; ski resorts; climate adaptation; Alpine tourism

## 1. Introduction

The influence of climate change on the tourism sector has been extensively discussed. Because of the outstanding role of the Alps as a leading tourism destination, many publications have focused on the impact of climate change on the Alps, specifically for winter tourism. Lack of snow in ski resorts was already discussed more than 20 years ago by Galloway [1], Koenig and Abegg [2] as an emerging threat for Alpine destinations. As tourists respond sensitively to lack of snow (Nicholls [3] and Uhlmann et al. [4]), the need for adaptation measures became increasingly clear. (Scott and McBoyle [5], Steiger and Mayer [6], Abegg et al. [7]). Snowmaking became a key technology for stabilizing snow levels, and it was installed in virtually every alpine destination (Rixen et al. [8], Evette et al. [9], Spandre et al. [10]). Strategic measures also focused on revenue management and pricing (Damm et al. [11], Malasevska and Haugom [12], Holmgren and McCracken [13]), finance and investments (Falk [14], Pickering and Buckley [15]).

In recent years, adaptation strategies have expanded from the level of ski resorts to the regional destination level (Jopp et al. [16], Hopkins [17]), for which climate scenarios are increasingly needed (Landauer et al. [18], Pröbstl et al. [19], Helgenberger [20]). In the search for strategies to stabilize and diversify destinations, studies have also investigated the impact of climate change on stakeholder perceptions (Wyss et al. [21], Trawöger [22]) and destination governance (Pechlaner et al. [23], Wyss [24]). Destinations are currently focusing on three main areas: understanding variance and change in current and future weather conditions better, developing tourism products that are more diversified and available in different seasons, and understanding visitor preferences and expectations.

However, tourist demand reponses to climate change remain a significant research gap (Gössling et al. [25]), specifically with regard to winter destination choice. Studies have investigated the importance of snow for winter sports tourism (Jacobsen [26]) and cross-country and downhill skiers' reactions to lack of snow (Unbehaun et al. [27], Pickering et al. [28], Pröbstl-Haider and Haider [29]). All of these studies were carried out with a focus on the influence of local weather. Yet it can be argued that motivations for winter holidays are far more complex, varying, for example, between alpine winter skiing destinations and destinations with multi-seasonal product offers (Bausch and Unseld [30]).

Destination choice has been the subject of tourism research since the early 1980s. In their 2005 review of the state of research on destination choice, Sirakaya and Woodside [31] showed that most models confirm the influence of cognitive and affective attributes forming destination images. Decrop [32] concluded that both cognitive and affective attributes play a major role when consumers define their destination choice parameters. In later models, dynamic feedback elements were added to consider previous destination experiences (Nilplub et al. [33]) as well as the importance of word of mouth from generic information sources (Crouch [34]). In addition to destination choice, destination loyalty was also identified as an important factor: Zang et al. [35] showed, for example, that the overall image of a destination has considerable influence on tourist loyalty.

In a recent study, Crouch et al. [34] (p. 574) analysed the correlation of vacation experiences, activity types and future experience preferences and related loyalty, concluding that there is considerable evidence for the stability of preferred experiences. Similar findings were presented by Hallmann et al. [36], who developed a model based on data from two Alpine winter sports destinations, confirming that destination image is multidimensional, with activities influencing the tourists's intention to revisit. In their discussion of tourist loyalty, McKercher, Denizci-Guillet and Ng [37] confirmed the importance of vertical and experimental loyalty in tourist choices, the latter referring to certain holiday styles. While McKercher et al. [37] found limited evidence of horizontal loyalty (loyalty to providers in the supply chain, such as hotels), Almeida-Santana and Moreno-Gil [38] showed that tourists are often loyal to a destination without being loyal to the tour operators, hotel companies or locations within a destination.

So far, no studies have analysed the influence of climate change on interrelated aspects of destination image, choice and visitor loyalty, for which the cognitive and affective attributes of climate change need to be better understood. This study focuses on winter tourism in the European Alps, where climate change is tangible (Gobiet et al. [39]).

## 2. Model and Research Methods

We hypothesize that direct and indirect cognitive or affective attributes of climate change have an influence on destination choice and loyalty. Direct cognitive attributes represent consumer knowledge based on personal experience, information provided by media, or word of mouth. For example, glacier melting or extremely late snowfall in the last few winters can be part of such knowledge. In contrast, direct affective attributes related to climate change are those creating or changing a feeling linked to a destination. Destinations can be perceived as pleasant or unpleasant, secure or risky, stressful or relaxing. Cognitive and affective attributes are strongly interlinked and these interrelations define the conative aspect of the destination choice [40]. Indirect effects of climate change caused by adaptation measures must also be considered. The installation and operation of snowmaking technology [41] can

change a tourist's destination image. A significant increase in ticket prices due to high investments and energy costs for snow production can be a further indirect effect. Indirect effects can become cognitive destination attributes, e.g., the knowledge about a newly constructed water basin, as well as affective, e.g., an unpleasant feeling caused by negative environmental perceptions.

In order to measure the potential cognitive and affective attributes of climate change, we used the destination choice model developed by Woodside and Lysonski [42], with amendments for feedback elements and uniqueness as described by Qu et al. [43]. Figure 1 shows the final model for our study:

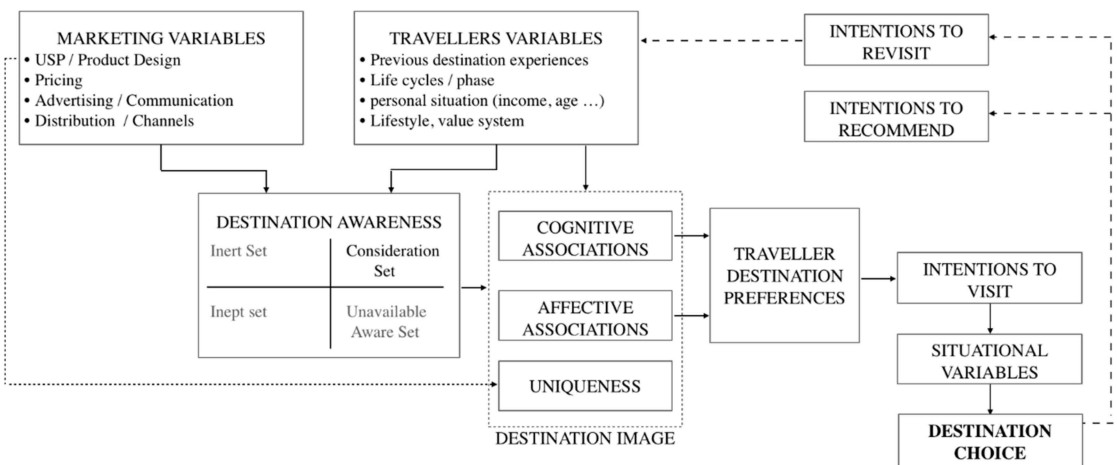

**Figure 1.** Destination choice model with choice sets, destination image and dynamic feedback loop.

A two-phase study design was applied for this research. We started with a qualitative study which identified relevant attributes of winter destination choice and also choice specifically related to climate change. The qualitative research results were then used to carry out a quantitative survey. Germany was chosen as source market for the analysis because it is the largest market for winter tourism in Europe and strategically important for many Alpine destinations. In addition, the competition in the German travel market among tour operators, low cost carriers, online distribution platforms and cruise ship companies is among the most intense in Europe.

The qualitative study was organized as a one-week online forum. QDC-Studio from Kernwert, Berlin was used as the technical basis. QDC-Studio offers a full responsive interface with a comprehensive qualitative data collection system. It allows a combination of forum group discussions with various individual tasks offered to the participants such as ranking of alternatives, selecting preferred attributes, or moving a pointer on a scale. The model from Figure 1 was released in a five-day working program as shown in Table 1. Forum participants had a daily workload of 15 to 20 min. To avoid creating a bias towards socially desirable responses, the topics and the wording of the forums were arranged without mentioning the term "climate change" during the first three days.

The participants were recruited from an online panel (NORSTAT Germany) using a quota plan. The sample structure represented German winter (Nov-Mar) travellers by age, sex, income and regional distribution. Half of the participants spent their last winter holiday in an Alpine destination. This split was necessary to identify general and also Alpine-specific destination choice attributes. A total of 81 participants took part in the online forum; 65 of them fitted the quota plan and finished all the tasks. These 65 participants received a maximum incentive of 20 Euros for taking part in all the activities during the week.

**Table 1.** Five days online forum structure of qualitative study based on the theoretical model.

| Day | Tasks Using Various Tools | Discussion in Forum/Chat | Part of Model |
|---|---|---|---|
| 1 | • Consideration and inept set today<br>• Motivation for having a winter holiday trip<br>• Changes of destinations during the last 10 years. | • What has changed during the last 10 years in my travel behaviour and why | • Destination awareness |
| 2 | • Most important expectations/wishes when organizing a holiday trip<br>• Cognitive and unique associations of favoured winter destination<br>• Affective associations | • What might change in the next 10 years in my travel behaviour and why | • Destination image: cognitive, affective and unique attributes—thereby climate change implicitly and not explicitly asked |
| 3 | • Role/importance of experiences, recommendations and information<br>• Value system/lifestyle of consumer using Sinus basic orientation<br>• Consumers personal situation in lifecycle and living conditions | • Role of weather and climate when choosing a destination | • Travellers' variablesIntentions to recommend or revisit<br>• Intentions to recommend or revisit<br>• Destination preferences |
| 4 | • Marketing variables, communication and advertising<br>• Pricing, distribution and booking | • 2 chat groups discussing main results of day 1–3 and topic climate change | • Marketing variables<br>• Climate change as impact factor to destination attributes explicitly asked |
| 5 | • Climate change<br>• Personal opinion in general and specifically<br>• Travelling instruments facing climate change: effectiveness and acceptance | • What does the term "sound environment/sound nature" mean?<br>• What role does it play when selecting a destination? | • Climate change awareness and attitude in context of travelling |

This qualitative study took place from 8 to 12 October 2018. The participants contributed to 1320 posts in discussion forums, 710 additional posts in two chat groups, and 1252 results from the 18 tasks using the QDC-Studio tools. A coding of the posts using traditional colour marking as well as coding software provided the basis for further qualitative data analysis. Four groups of young researchers did a comparative text reading and result synthesis. A group discussion among all the young researchers and two of the authors, who read and analysed all the text and tasks, led to the results presented here. This approach followed the grounded theory method described by Glaser and Strauss [44]. Most of the 18 tasks delivered data as preferences, rankings or scale values. These data were analysed using SPSS 25. Although the sample size of 65 cases was large enough to use analytical statistical methods, the authors worked solely with descriptive methods due to the qualitative nature of the survey. The main purpose of the qualitative study was to provide a conceptual framework for the subsequent quantitative survey with a focus on the role and impact of climate change in the winter destination choice process and horizontal destination loyalty [37] as defined by McKercher.

The subsequent quantitative survey concentrated on German travellers who had made regular winter holiday trips to an Alpine destination during the last five years. Within this group, a representative sample based on the quotas from the German travel analysis [45] for Alpine winter holidays was generated, using gender, age, family status, education, number of children in the household and income. To analyse whether climate change triggers a change of winter destination in terms of horizontal loyalty, we identified three different groups: 'Alpine repeaters' for the coming winter season (loyal), 'leavers' visiting non-Alpine destinations in the coming winter season (disloyal), and a group of people either undecided or who will not travel at all in the coming winter season, the 'undecided'. Figure 2 shows the split of the total sample into the three sub-groups.

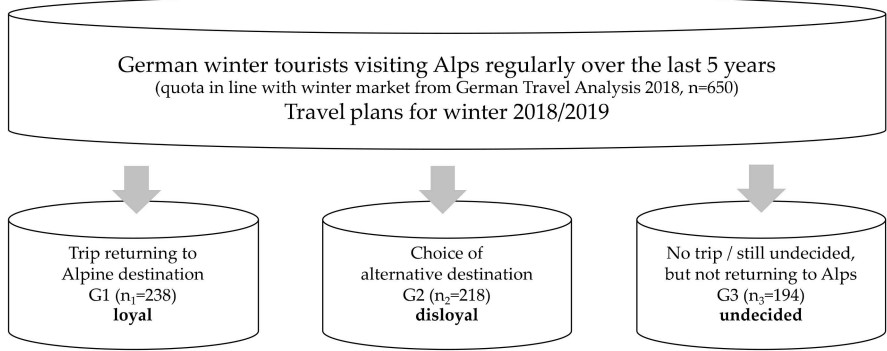

**Figure 2.** Sample design of quantitative study.

The survey participants were again recruited from the German NORSTAT panel. A fully structured questionnaire was developed, pre-tested and implemented to collect the data. The standards described by Dolnicar [46] were applied to develop the items and associated scales. It was possible for the participants to give "don't know" responses, and a binary (yes/no) scale was used for "either/or" questions in order to improve the data quality as demonstrated by Dolnicar and Grün [47]. The design of the quantitative study was based on the three groups described in Figure 2, allowing comparison and analysis of factors as outlined in Figure 3. Various methods were used to identify factors with significant influence on the decision to re-visit an alpine winter holiday region (G1 = group 1, loyal), to choose a new winter destination located outside the Alps (G2 = group 2, disloyal) or to stop travelling or remain undecided until November (G3 = group 3, undecided) (Table 2). The definition of horizontal loyalty follows Almeida-Santana and Moreno-Gil [38], and the Alps are characterized as one winter tourism destination.

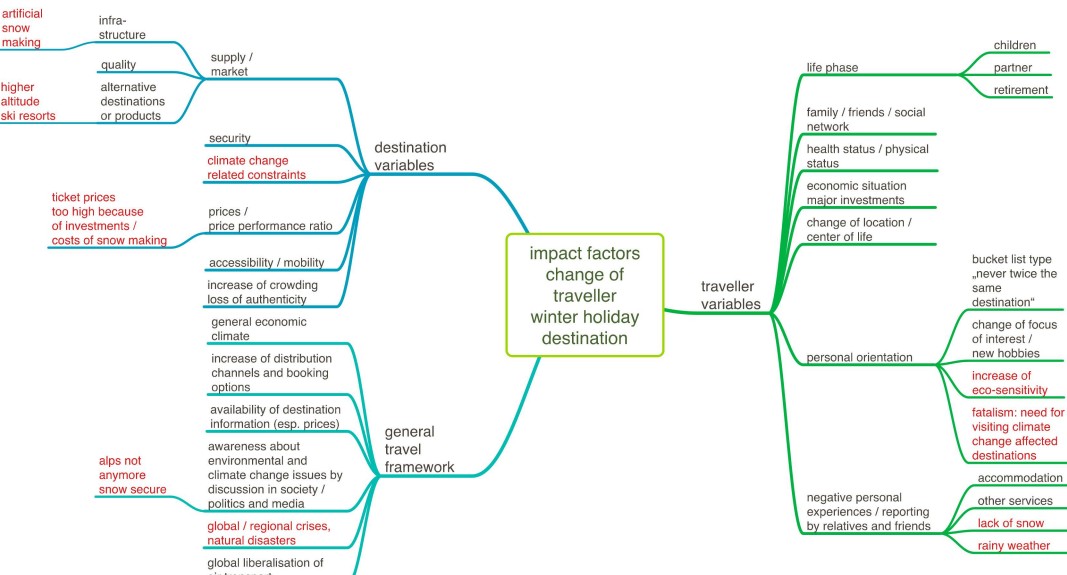

**Figure 3.** Synthesis of factors with impact on destination choice—climate change highlighted in red.

**Table 2.** Questionnaire structure with blocks of factors, aspects and scales.

| Block | Aspects |
|---|---|
| *Block name* <br> *[number of items]* | *[number of items if more than one] (scales: b: binary, n-n: nominal with n categories, o-n: ordinal with n ranks, l-n: Likert with n points, c: cardinal numbers)* |
| Traveller variables (18) | Changes in life during the last year: children [4], partnership/own household [3], job [2] (b), new place of residence [2], health status/care [3], economic situation [3] all scaled (b) |
| Destination variables (57) | Place of stay and hosts [21], typical winter holiday experiences [16], landscape, nature and culture [11] all scaled (a) importance (b) (b) satisfaction last stay (l5) awareness of climate change in the Alps [5] (b) and the last Alpine winter destination [5] (b) |
| General travel framework (13) | New destinations and offers [2], price transparency and budget [2], information channels destinations [2], booking channel and date [2] information sources climate change [5] all scaled (b) |

Based on the results of the qualitative study, questions and items for a fully structured questionnaire were developed. Table 2 shows the questionnaire structure:

The field phase was held 12–17 November 2018. In total, 650 (n) questionnaires were fully completed, equally split among the three sub-groups ($n_1$ = 238, $n_2$ = 218, $n_3$ = 194) and fitting the quota plan adequately. The average time taken to fill in the questionnaire was about 16 min, and this was remunerated with 8 euros. The statistics were analysed with SPSS25.

## 3. Results

### 3.1. The Role of Climate Change among Other Destination Choice Factors: Results from the Qualitative Study

The one-week online forum produced results covering all components of the model from Figure 1. The findings dealing with marketing and traveller variables were used to derive the items for the quantitative study and will be presented later. The focus is on climate change as a factor influencing destination choice and thereby destination loyalty.

We started by analysing the destination choice set. Two aspects were examined in order to identify climate change as an influential factor outside the marketing and traveller variables in the model. First, based on the dimensions set up by Bausch and Unseld [30], we looked at the personal motivations for a winter holiday trip. Participants had to assign each dimension to one of three options: 'that's me', 'that's only partially me', 'that's not me'. A narrative approach was then used to find general factors that play a role when thinking about destinations for a winter holiday trip. The participants had to answer the following questions: "Reflecting on your winter holiday trips in the last ten years, what has changed? Have you changed the period, the duration, or the organization of your trips? Have any aspects of your personal situation changed? When you are planning a trip, are there any aspects that are more important or less important today than in the past?"

Four aspects of the personal motives for taking a winter holiday trip emerged as very important to nearly a third of the respondents: bridging the long period until the next summer, finding less crowded destinations, finding offers at lower prices and having more light and sun than at home. Only a quarter of the participants put doing winter sports or experiencing real winter with snow and low temperatures in the category "that's me". Two thirds of the participants put leaving the dull autumn, enjoying winter atmosphere and experiencing warmth in the category "that's partially me". This shows that weather plays an important role in destination choice. Therefore, perceptions of longer-term climate change or observed weather changes in the destination might influence whether travellers revisit the Alps ("loyal") or not ("disloyal" and "undecided").

Our analysis of the posts of the narrative part ("what has changed?") identified seven main topics: see Table 3.

**Table 3.** General factors influencing changes of destination choice.

| Topic | Aspects |
|---|---|
| Range of available travel products | New destinations (globalization, long haul destinations), more variety (new types of holiday trips), fast growing cruise market and new destinations (me-too product, negative ecological impact, mass-tourism) |
| Information and booking | New and additional information/booking by internet (market overview, comparability, online booking, faster processes, last minute decision possible, easy to use), change of services by travel agencies (fewer agencies, more specialized, higher reliability) |
| Price level/costs | Greater spread of price-range from exclusive and very expensive to very cheap/low cost market, higher range of cheap products especially flights and all-inclusive |
| General changes of society/world | Expectation level of consumers in tourism increased, new guest groups from other countries appearing in destinations (esp. eastern European countries/Russia), safety/terrorism a new aspect of destination choice |
| Weather | Before travelling: short term decision based on reliable forecasts: last minute booking of new/other destinations/cancelling of a booking. While travelling: changing location during the trip (esp. in case of rain in mountains switching to higher regions) |
| Use of the internet during trip | Availability for work during holidays, information search at destination, ad hoc booking of services at location, less "real" experiences |
| Personal situation | Family (birth of children/empty nesters), partnership (new/ended), income (better or weaker, retirement), significant change of health status (improvement/decline) |

The first time any participants explicitly mentioned climate change when recalling their holidays was when they were thinking about their past experience of winter weather Four out of more than seventy participants associated weather phenomena with climate change, including lack of snow, rainfall during the winter or extreme snowfall causing avalanches.

In some cases, travellers' characters were identified as a further reason for changing destination. Five participants mentioned that they would never travel to the same place twice because they are interested in seeing different parts of the world. In these cases, destinations will never be able to create loyalty as basis for repeat visits [48].

The next step was the analysis of the cognitive, unique and affective associations of the destination image. First, the participants were asked in a narrative format to name their favoured winter destination for the upcoming winter season and then to describe it. They also had to explain what makes this destination unique. The most frequently described cognitive aspects of the destinations were natural features, landscape and culture, the politeness of the locals, and local food. In addition, for warm destinations, warm, sunny weather and beach-related activities were mentioned. Alpine destinations were frequently connected with winter atmosphere and ski resorts. For all types of destinations, weather was associated with positive pictures like sunny seaside, white mountains in bright light, fresh air and blue sky. Destination uniqueness is linked with outstanding natural or cultural features or unique experiences when meeting the locals. The posts therefore represent an ex ante positive projection of a perfect stay in the destination. This explains why climate change and its potential negative impacts are not mentioned by the participants as part of cognitive destination attributes, and why they do not feature in the affective part of the destination image projection.

At the end of the third day of the online forum, the participants were explicitly asked to discuss the role of weather and climate when choosing a destination. The term climate was used by the moderators but not the term climate change. The analysis of the 110 posts registered only four direct mentions of climate change and twelve indirect references by describing typical climate change related weather unusual conditions. Three direct comments stated that in the future skiing in the destination

might no longer be possible due to climate change. One reference was made to extreme heat in a long-haul destination and the potential need to shift the travelling period. Climate change effects described were: lack of snow (3), unreliable snowfall (3), late snowfall (2), rainfall (2) and extreme heat (2). This could imply that some travellers are already aware of climate change and its effects.

To obtain a more detailed picture of some aspects which showed up in the forum during the first three days, two one-hour chat groups with 11 and 13 study participants took place on day four. The chats covered five thematic blocks, one of which was climate change. During the chats the following additional aspects were related to climate change:

- Some participants expressed concern over their flight-related carbon footprint. They underlined the need to change their travel behaviour and to avoid air travel in the future.
- A further reaction was fatalism. The argument was that climate change is a fact we have to live with, and therefore places and attractions endangered by climate change should be visited as soon as possible, while they are still there and accessible.
- More general remarks referred to an increasing risk from natural hazards and new restrictions for guests.
- A final thematic block was the increasingly unreliable snowfall. One aspect was adaptation by either machine-made snow or by moving to regions at higher altitude. The majority of participants in this group were sceptical about a further technical upgrading of ski resorts because of the negative environmental impact. A second aspect was the increasing prices for tickets because of higher investments, increased snow production and intensified slope management.

Finally, we identified three groups of impact factors that play a role during the destination choice and might have an influence on destination loyalty: traveller variables, destination variables and the general travel framework conditions. Factors marked in red in Figure 3 show the potential role of climate change.

### 3.2. Quantitative Study

First, the three blocks from Table 2 were analysed using Principal Component Analysis (PCA) in combination with varimax rotation. As shown in Table 4, the set of variables from the first block describing changes in life during the last year were grouped into 5 factors with eigenvalues above 1.0 explaining in total 54.9% of the total variance. KMO (Kaiser-Meyer-Olkin measure of sampling adequacy) was 0.835, indicating an adequate sampling. The factors describe (share of explained variance in parentheses): F1 family structure (26.0%), F2 job and housing (9.0%), F3 partnership and household (7.9%), F4 improvements in health and economic situation (6.2%) and F5 constraints in health (5.9%), personal and economic situation. The influence of these five factors, as well as each of the 18 single variables, was tested by pairwise *t*-tests. Comparing the groups shows which factors or variables might influence a change from Alpine winter destinations to other destinations (G1–G2) or cause travellers to give up winter holidays in the Alps (G1–G3). At the factor level, the G1–G2 comparison only shows a significant influence for factor F5. A decline in health or personal and economic situation can lead winter season travellers to look for alternative destinations. A new job also forces people to change their winter destination because it brings uncertainty for holiday planning, and therefore causes a temporary change in behaviour. The single variables in factor F5 show that health plays the most important role. Considering that Alpine winter holidays are often linked to winter sports, this result is not surprising, because sports require good health as well as the financial resources for tickets and equipment.

**Table 4.** Traveller variables analysis results.

| Traveller Variables Personal Situation | PCA Rotated Factor Loadings Varimax | | | | | Pairwise *t*-test Repeater—Non Repeater | | | |
|---|---|---|---|---|---|---|---|---|---|
| | Factor No. (Eigenvalue > 1.0) | | | | | G1–G2 | | G1–G3 | |
| | F1 | F2 | F3 | F4 | F5 | *t* | Sig. [c] | *t* | Sig. [c] |
| birth of child | 0.65 [a] | | | | | | | | |
| school enrolment of child | 0.69 [a] | | | | | | | 2.153 | 0.016 ** |
| last child finished school | 0.71 [a] | | | | | | | 1.811 | 0.036 ** |
| first time holidays without kids | 0.58 [a] | | | | | | | 2.260 | 0.012 ** |
| retirement | 0.46 [b] | | | | 0.39 [b] | | | | |
| new partnership | | | 0.77 [a] | | | | | 1.847 | 0.033 ** |
| end of partnership/loss of partner | | | 0.76 [a] | | | | | | |
| first-time own household | | 0.46 [b] | 0.50 [b] | | | | | 2.914 | 0.002 ** |
| new job | | 0.49 [b] | | | | −2.979 | 0.002 ** | | |
| loss of job/partially unemployed in 2018 | | 0.39 [b] | 0.35 [b] | | | | | 2.153 | 0.016 ** |
| removal to new flat/house | | 0.82 [a] | | | | | | | |
| removal to new residence | | 0.82 [a] | | | | | | | |
| new factors in family (e.g., care for relatives) | | | | | 0.41 [b] | | | | |
| significant improvement of health | | | | 0.67 [a] | | | | 1.363 | 0.087 * |
| sign. decline of health | | | | | 0.83 [a] | −1.852 | 0.033 ** | | |
| sign. improvement of economic situation | | | | 0.79 [a] | | 1.625 | 0.053 * | 1.487 | 0.069 * |
| sign. decline of economic situation | | | | | 0.69 [a] | | | 1.753 | 0.040 ** |
| higher investments (e.g., new car/flat) | | | | 0.57 [a] | | | | 2.824 | 0.003 ** |
| G1–G2    *t*-value | 0.047 | −0.636 | −0.031 | 0.886 | −1.675 | | | | |
|    sig. 1-tailed | 0.482 | 0.263 | 0.488 | 0.188 | 0.048 ** | | | | |
| G1–G3    *t*-value | 2.172 | 0.567 | 0.918 | 2.073 | −0.207 | | | | |
|    sig. 1-tailed | 0.015 ** | 0.286 | 0.180 | 0.020 ** | 0.418 | | | | |

[a] main part of factor loading on one factor, loading greater than 0.5. [b] factor loading split to several factors or loading not greater than 0.5. [c] significance calculated as 1-tailed *t*-test; ** if smaller than 0.05, * if smaller than 0.1 sds.

Conversely, the G1–G3 comparison shows that both the family factor and an improvement in health and economic status lead people to return to Alpine winter destinations. Families with schoolchildren are tied to school holidays, the most expensive seasons, whereas travellers who are not restricted to school holidays find attractive offers for mid-season winter trips at reasonable prices. This explains why the family factor variables "last child finished school" and "first-time holidays without kids" had a positive influence on the choice of the Alps. At first sight it seems contradictory that the variable "school enrolment of a child" should lead to guests returning to the Alps. A possible explanation is that for a family, warm winter destinations are just as expensive during winter school breaks as one-week vacation in the Alps. There is therefore no major price difference between Alpine and non-Alpine destinations which improves the competitiveness of the Alps. Although F3 is not significant at the factor level, three variables have a positive influence on guests returning to Alpine destinations: a new partnership, a first household of one's own, and job loss or partial unemployment during the year 2018. These variables describe a phase of personal change that might cause people to hang on to some traditions like a winter holiday trip to the Alps.

All the items that measure the destination attributes and guest experiences were derived from the qualitative study along the 'customer journey' (Yalchin [49]) by analysing the posts from the answers to the online forum questions "When you think about a winter holiday in the Alps, what comes to mind? What features do you expect to find in the destination? What type of experiences do you hope to make?". This finally led to 48 items. In the quantitative study, the interviewees were first asked which of these 48 items were important to them when choosing an Alpine winter destination. In a second step, they had to rate the importance of the items they had selected in the first round. The rating was done on a 5-point Likert scale measuring the level of fulfilment of their expectations during their last stay: much more (1), more (2), as expected (3), less (4) and much less (5). The value 3 "as expected" was inserted for the items that an interviewee had not selected as important. The rationale behind this addition is simple: when a destination attribute has no importance for a guest, it will not play a role in destination choice and therefore the fulfilment level of the expectation is neutral. Using these data with the addition of the missing values for a PCA, the seven factors shown in Table 5 were identified with a total explained variance of 56.1% at eigenvalues above 1.0 and the absolute factor scores of the 48 items above 0.35 after varimax rotation. KMO was 0.967, indicating an adequate sampling. The factors F1 to F7 (share of explained variance of rotated factors in brackets) are sorted by declining factor loadings and therefore cumulative importance for the guests.

The results show that winter sports are not the dominant factor when choosing a winter destination. Guests weighted the basic destination quality (F1) and the general performance and prices of the available infrastructure and services (F2) much more heavily. The potential for Alpine winter experiences (F3) framing the activities during their stay (F4) comes in third place. Winter sports are one important activity among many others. Only 67.4% of all respondents state that winter sports are important to them. This implies that one third of the people who have a regular winter holiday trip to an Alpine destination have other preferences than just winter sports. Furthermore, the pure nature experience is listed as a separate factor (F5).

An in-depth analysis of all 48 items followed to search for significant disparities among the groups of loyal guests (G1) and those who will not visit the Alpine destination again (G2 disloyal guests, G3 undecided guests), hereafter called loyals (G1), disloyals (G2) and undecided (G3). To compare the Likert-scaled ratings, a non-parametric Mann-Whitney U-test was used. As this test does not indicate which of two groups has a higher ranking, the *t*-values of a two-sample *t*-test for independent samples was used as well. Table 6 summarizes the results.

**Table 5.** Destination variables fulfilment of expectations PCA results.

| Factor | Characteristics | Items with Absolute Factor Loadings above 0.35 |
|---|---|---|
| F1 (12.2%) | Basic destination quality and atmosphere | Accessibility, atmosphere for recreation, typical Alpine scenery, cosy authentic accommodation, accommodation comfort, excellent food/board, attractive ski resort/cross-country tracks, cheerfulness of hosts, authenticity of hosts, cordiality in destination, enjoying winter nature, mountain panorama |
| F2 (11.3%) | Performance and prices of infrastructure/services | Accessibility, tourist information, accommodation price, food/meal prices, attractive ski resort/cross-country tracks, high-performance ski lifts, snow security/machine-made snow, ski lift prices, supply ski-/snowboard schools and rental, prices ski-/snowboard schools and rental, shopping facilities food/convenience goods, shopping, strolling around, skiers' respect for others on the slopes, sufficient space on slopes |
| F3 (10.5%) | Alpine winter experience | Reliable snowfall/machine-made snow, enjoying winter nature, walks/hiking, typical Alpine food and drink, going out into the cold clear winter air, snow-covered landscape and forest, unspoiled pristine landscape, mountain panorama, experiencing snowfall, silent areas without noise/traffic, frozen lakes and streams, icicles, solitary places/trails/tours, protected areas, plenty of sun, glistening light |
| F4 (8.3%) | Activities during stay | Visiting mountain huts/restaurants, winter sports (Alpine or Nordic), ski touring/snowshoeing, sledging/tobogganing, après-ski/entertainment/disco, wellness/getting pampered, ice-skating, playing together in snow, Alpine winter traditions/Christmas |
| F5 (5.6%) | Pure nature experience potential | Unspoiled pristine landscape, observing wildlife in winter, silent areas without noise/traffic, solitary places/trails/tours, protected areas |
| F6 (4.8%) | Hedonic experience potential | Shopping, strolling around, visiting spa/thermal springs, wellness/getting pampered, sunbathing, plenty of sun, glistening light |
| F7 (3.4%) | Price level core services | Accommodation price, food/meals price, cable car/lift prices, après-ski/entertainment/disco. |

The most significant differences between G1 and G2 are in the attributes linked to skiing. Loyals rate winter sports as important (G1 76%:G2 65%) and therefore attach significantly higher importance to attractive ski resorts or cross country tracks (75%:67%), high performance ski lifts (74%:64%), sufficient space on slopes (81%:69%), skiers' respect for others on the slopes (79%:74%) and après ski and disco (50%:42%). In connection with climate change, reliable snow conditions and machine-made snow (80%:72%) are the only items of higher importance to loyals than to disloyals. A higher share of loyals expect typical Alpine scenery (G1 76%:G2 70%), typical Alpine food and drink (79%:70%), Alpine winter traditions (65%:56%) and meeting locals (74%:65%). There are only two attributes in which disloyals (G2) expect more than loyals (G1): protected areas (G1 66%:G2 76%) and accommodation price (G1 76%:G2 83%). This suggests that disloyals (G2) are more eco-sensitive and more price-sensitive because they attach less importance to winter sports. 90% of non-skiers expect to enjoy real winter atmosphere, so snow-covered landscape and forest scenery are of higher relevance for their stay (87%:92%). Non-skiers do not accept higher prices for accommodation located near the ski-resort.

**Table 6.** Destination variables expectations and experiences groups G1 to G3.

| Destination variables | Expectations | | | | | | | Experiences Last Winter Holiday Trip in the Alps | | | | | | | | |
| | Expect. Level | | | Pairwise *t*-Test Repeater—Non Repeater | | | | Unsatisfied Last Stay | | | G1–G2 | | | G1–G3 | | |
| | | | | G1–G2 | | G1–G3 | | | | | *t*-test | MW—U-test [b] | | *t*-test | MW—U-test [b] | |
| | Ø G1 | Ø G2 | Ø G3 | t | Sig. [a] | t | Sig. [a] | % G1 | % G2 | % G3 | t | std. U [c] | Sig. [d] | t | std. U [c] | Sig. [d] |
|---|---|---|---|---|---|---|---|---|---|---|---|---|---|---|---|---|
| accessibility | 0.82 | 0.79 | 0.83 | | | | | 2.1 | 1.7 | 5.6 | | | | −2.543 | 2.396 | 0.017 ** |
| tourist information | 0.59 | 0.58 | 0.57 | | | | | 3.5 | 4.8 | 7.3 | | | | −2.138 | 2.204 | 0.028 ** |
| atmosphere for recreation | 0.90 | 0.88 | 0.89 | | | | | 2.3 | 6.3 | 3.5 | | | | −2.767 | 2.663 | 0.008 ** |
| typical Alpine scenery | 0.76 | 0.70 | 0.69 | 1.416 | 0.079 * | 1.522 | 0.065 * | 3.9 | 4.6 | 4.5 | | | | −2.246 | 2.294 | 0.022 ** |
| cosy authentic accommodation | 0.84 | 0.82 | 0.79 | | | | | 3.0 | 2.8 | 5.2 | | | | −1.855 | 1.848 | 0.065 * |
| accommodation comfort | 0.82 | 0.83 | 0.84 | | | | | 2.0 | 3.3 | 5.6 | | | | −3.079 | 2.949 | 0.003 ** |
| accommodation price | 0.76 | 0.83 | 0.87 | −1.947 | 0.026 ** | −3.038 | 0.002 ** | 5.0 | 10.5 | 6.5 | −2.224 | 2.234 | 0.026 ** | −4.007 | 4.156 | 0.000 ** |
| excellent food/board | 0.91 | 0.91 | 0.90 | | | | | 3.2 | 3.5 | 3.4 | | | | | | |
| food/meals price | 0.77 | 0.81 | 0.79 | | | | | 7.6 | 11.4 | 11.8 | | | | −2.072 | 2.215 | 0.027 ** |
| attractive ski resort/cross-country tracks | 0.75 | 0.67 | 0.70 | 1.946 | 0.026 ** | | | 1.7 | 4.1 | 3.7 | | | | −3.716 | 3.979 | 0.000 ** |
| high-performance ski lifts | 0.74 | 0.64 | 0.64 | 2.361 | 0.010 ** | 2.361 | 0.010 ** | 4.0 | 5.0 | 4.8 | −1.418 | 1.689 | 0.092 * | −3.479 | 3.631 | 0.000 ** |
| reliable snow conditions/machine-made snow production | 0.80 | 0.72 | 0.73 | 2.071 | 0.020 ** | 1.738 | 0.042 ** | 10.5 | 9.6 | 16.2 | | | | −3.927 | 3.977 | 0.000 ** |
| ski lift prices | 0.66 | 0.63 | 0.69 | | | | | 10.2 | 17.5 | 20.9 | −2.464 | 2.482 | 0.013 ** | −4.772 | 4.576 | 0.000 ** |
| supply ski-/snowboard schools and rental | 0.54 | 0.52 | 0.40 | | | 2.918 | 0.002 ** | 2.3 | 10.5 | 14.1 | −1.876 | 1.975 | 0.048 ** | −4.041 | 3.704 | 0.000 ** |
| prices ski-/snowboard schools and rental | 0.54 | 0.53 | 0.46 | | | 1.529 | 0.064 * | 7.8 | 15.5 | 14.4 | | | | −3.065 | 3.118 | 0.002 ** |
| cheerfulness of hosts | 0.92 | 0.91 | 0.90 | | | | | 1.4 | 4.0 | 2.9 | | | | −2.660 | 2.748 | 0.006 ** |
| authenticity of hosts | 0.68 | 0.61 | 0.59 | 1.774 | 0.039 ** | 2.102 | 0.018 ** | 2.5 | 1.5 | 5.3 | | | | −2.244 | 2.559 | 0.011 ** |
| cordiality if hosts in destination | 0.91 | 0.88 | 0.92 | | | | | 2.8 | 1.0 | 4.5 | | | | | | |
| shopping facilities food/convenience goods | 0.70 | 0.71 | 0.66 | | | | | 6.6 | 4.5 | 8.6 | | | | −2.422 | 2.608 | 0.009 ** |
| shopping, strolling around | 0.53 | 0.49 | 0.46 | | | | | 6.4 | 6.6 | 12.2 | | | | −2.421 | 2.306 | 0.021 ** |
| skiers respect for others on the slopes | 0.79 | 0.74 | 0.71 | 1.404 | 0.081 * | 2.000 | 0.023 ** | 12.7 | 21.7 | 29.0 | −2.161 | 2.135 | 0.033 ** | −4.511 | 4.551 | 0.000 ** |
| sufficient space on slopes | 0.81 | 0.69 | 0.68 | 2.952 | 0.002 ** | 3.154 | 0.001 ** | 6.7 | 20.5 | 15.9 | −2.035 | 2.065 | 0.039 ** | −3.659 | 3.479 | 0.001 ** |
| visit to mountain huts/restaurants | 0.78 | 0.78 | 0.78 | | | | | 4.3 | 8.9 | 8.6 | −2.005 | 2.168 | 0.030 ** | −3.182 | 3.059 | 0.002 ** |
| winter sports (Alpine or Nordic) | 0.76 | 0.65 | 0.60 | 2.468 | 0.007 ** | 3.569 | 0.000 ** | 2.8 | 4.9 | 2.6 | | | | −1.644 | 1.846 | 0.065 * |
| ski touring/snowshoeing | 0.53 | 0.52 | 0.46 | | | 1.354 | 0.088 * | 5.6 | 7.1 | 11.1 | | | | −2.535 | 2.986 | 0.003 ** |
| enjoying winter nature | 0.89 | 0.90 | 0.92 | | | | | 5.7 | 6.6 | 6.2 | | | | | | |
| walks/hiking | 0.81 | 0.82 | 0.82 | | | | | 3.1 | 3.4 | 5.0 | | | | −2.566 | 2.464 | 0.014 ** |
| Alpine typical food and beverage | 0.79 | 0.70 | 0.74 | 2.170 | 0.016 ** | | | 4.3 | 6.5 | 6.9 | | | | −2.692 | 2.462 | 0.014 ** |
| going out into the cold clear winter air | 0.88 | 0.83 | 0.90 | 1.459 | 0.073 * | | | 3.3 | 1.6 | 2.9 | | | | −1.780 | 1.787 | 0.074 * |
| visit to SPA/thermal springs | 0.60 | 0.60 | 0.56 | | | | | 4.9 | 12.2 | 15.7 | | | | −2.142 | 2.290 | 0.022 ** |
| sledging/tobogganing | 0.61 | 0.58 | 0.52 | | | 1.961 | 0.026 ** | 4.8 | 15.0 | 10.0 | | | | −3.035 | 2.600 | 0.009 ** |
| après-ski/entertainment/disco | 0.50 | 0.42 | 0.35 | 1.670 | 0.048 ** | 3.261 | 0.001 ** | 6.7 | 14.1 | 11.9 | | | | | | |

**Table 6.** *Cont.*

| | Expectations | | | | | | | Experiences Last Winter Holiday Trip in the Alps | | | | | | | | |
|---|---|---|---|---|---|---|---|---|---|---|---|---|---|---|---|---|
| | Expect. Level | | | Pairwise *t*-Test Repeater—Non Repeater | | | | Unsatisfied Last Stay | | | G1–G2 | | | G1–G3 | | |
| wellness/getting pampered | 0.71 | 0.72 | 0.65 | | | | | 6.5 | 8.3 | 11.1 | | | | −2.378 | 2.373 | 0.018 ** |
| sunbathing | 0.51 | 0.52 | 0.50 | | | | | 9.0 | 10.5 | 9.3 | | | | −1.934 | 1.874 | 0.061 * |
| ice-skating | 0.43 | 0.38 | 0.25 | | | 4.086 | 0.000 ** | 9.7 | 14.6 | 10.4 | −2.784 | 2.629 | 0.009 ** | −3.014 | 2.890 | 0.004 ** |
| playing together in snow | 0.64 | 0.67 | 0.64 | | | | | 8.5 | 12.9 | 10.5 | | | | −1.968 | 1.848 | 0.065 * |
| Alpine winter traditions/Christmas | 0.65 | 0.56 | 0.48 | 1.812 | 0.036 ** | 3.436 | 0.001 ** | 3.9 | 9.8 | 10.6 | | | | −2.783 | 2.764 | 0.006 ** |
| meeting locals | 0.74 | 0.65 | 0.63 | 2.154 | 0.016 ** | 2.484 | 0.007 ** | 7.4 | 11.3 | 10.7 | −1.627 | 1.686 | 0.092 * | −3.077 | 3.119 | 0.001 ** |
| snow-covered landscape and forest | 0.87 | 0.87 | 0.92 | | | −1.712 | 0.044 ** | 9.7 | 9.5 | 12.9 | | | | −2.738 | 2.676 | 0.007 ** |
| unspoilt pristine landscape | 0.85 | 0.85 | 0.88 | | | | | 6.9 | 14.1 | 15.2 | −1.647 | 1.651 | 0.099 * | −3.169 | 3.318 | 0.001 ** |
| mountain panorama | 0.89 | 0.86 | 0.90 | | | | | 4.2 | 4.8 | 2.9 | | | | | | |
| observing wildlife in winter | 0.64 | 0.69 | 0.65 | | | | | 17.8 | 25.3 | 22.2 | −2.065 | 2.267 | 0.023 ** | −3.161 | 3.342 | 0.001 ** |
| experiencing snowfall | 0.75 | 0.78 | 0.79 | | | | | 12.3 | 21.2 | 23.4 | −1.596 | 1.688 | 0.091 * | −4.209 | 3.871 | 0.000 ** |
| silent areas without noise/traffic | 0.86 | 0.83 | 0.86 | | | | | 8.3 | 12.8 | 15.7 | | | | −3.071 | 3.435 | 0.001 ** |
| frozen lakes and streams, icicles | 0.73 | 0.76 | 0.75 | | | | | 9.8 | 17.0 | 18.5 | | | | −3.263 | 3.267 | 0.001 ** |
| solitary places/trails/tours | 0.65 | 0.67 | 0.65 | | | | | 8.4 | 13.6 | 17.3 | | | | −2.477 | 2.489 | 0.013 ** |
| protected areas | 0.66 | 0.76 | 0.73 | −2.399 | 0.009 ** | −1.501 | 0.067 * | 6.4 | 15.1 | 14.2 | −1.860 | 1.791 | 0.073 * | −3.238 | 3.146 | 0.002 ** |
| plenty of sun, glistening light | 0.77 | 0.80 | 0.76 | | | | | 9.8 | 7.5 | 8.8 | | | | −1.871 | 1.955 | 0.051 * |

[a] significance calculated as 1-tailed *t*-test; ** if smaller the 0.05, * if smaller than 0.1. [b] Mann-Whitney U-test. [c] standardized test statistic Man-Whitney U-test. [d] significance Mann-Whitney U-test two-tailed.

Comparing the groups G1 and G3, very similar results for the above attributes can be found. The differences between G1 and G3 tend to be even greater than between G1 and G2, due to the fact that in G3 only 60% indicate that winter sports are important for them, as against 65% in G2. It follows that all the attributes linked to winter sports are weighted lowest in G3. Significant differences between G1 and G3 are found for ski- or snowboard schools and rental with regard to supply (G1 54%:G3 40%) and prices (54%:46%), for ski touring and snow-shoeing (53%:46%), sledging (61%:52%) and ice-skating (43%:25%). G2 and G3, although not significantly different from each other, tend to have higher expectations regarding the cosiness, authenticity and naturalness of the destination.

In the analysis of the level of fulfilment of expectations during guests' last stay in their Alpine destination, G3 shows a significantly lower level in nearly all attributes. Being less focused on winter sports, G3 do not often seem to find what they expect in today's Alpine destinations. Even the group of G3 guests who rated winter sports as an important activity registered a high level of dissatisfaction with their last stay: skiers' respect for others on the slopes (29%), ski lift prices (21%), snow security and machine-made snow (16%), and sufficient space on slopes (16%). There is likewise a high proportion of dissatisfied G3 guests in the group expecting to experience nature: experiencing snowfall (23%), observing wildlife in winter (22%), frozen lakes and streams (19%), solitary places, trails and tours (17%), silent areas without noise and traffic (16%), and unspoiled pristine landscape (15%).

A closer look at the significant differences between G1 and G2 reveals two smaller blocks. Firstly, some ski sports related attributes show a higher dissatisfaction level in group G2. Particularly noticeable are: skiers' respect for others on the slopes (G1 13%:G2 22%), sufficient space on slopes (G17%:G2 20%), ski lift prices (G1 10%:G2 18%), and prices for courses and rental (G1 8%:G2 15%). Secondly, as in G3, nature-related attributes show significant differences: observing wildlife in winter (G1 17%:G2 25%), experiencing snowfall (G1 12%:G2 21%), protected areas (G1 6%:G2 15%) and unspoiled pristine landscape (G1:G2 14%).

The qualitative study also identified a third block of influencing variables: the general travel framework. All related aspects listed in Table 2 were transferred into statements the interviewees could agree or disagree on. For example, a participant in the online forum stated: "today there are an enormous number of winter travel options and therefore I study them more than I used to and think about new and alternative options." Based on this result, the statement "I study the large choice of winter destinations carefully" was developed. Figure 4 shows all 8 statements, the level of agreement in each of the groups G1 to G3 and the results of a two-sample *t*-test for independent samples.

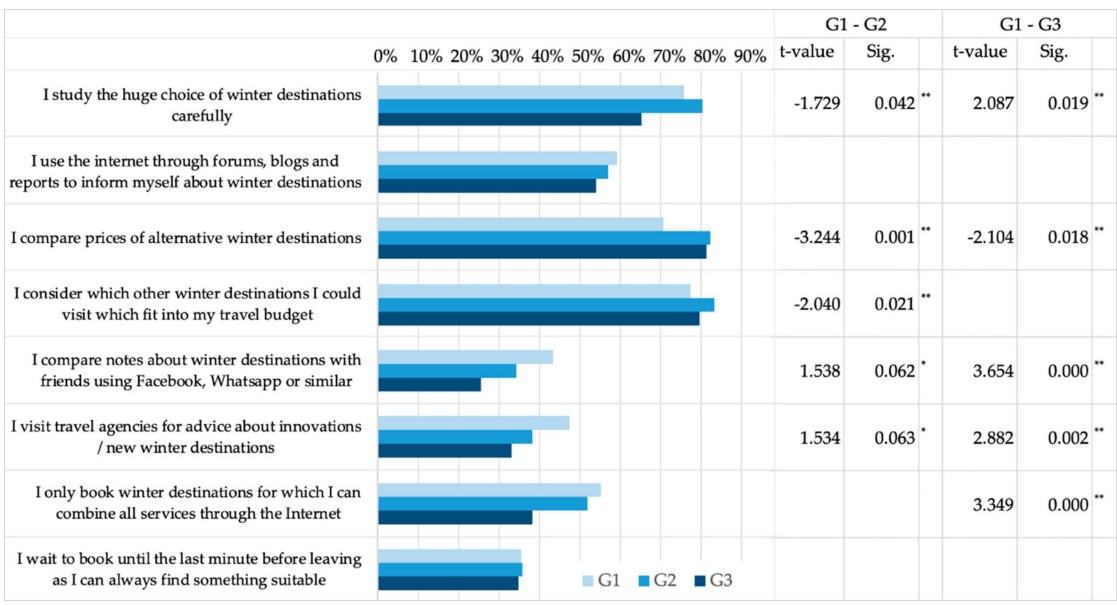

**Figure 4.** Information and booking of winter holiday trips.

The increase of winter destinations and products accompanied by the availability of information on the internet has different impacts on the destination choice in the three groups. Disloyal travellers from G2 show the highest confirmation of the statement "I study the large choice of winter destinations carefully". They also show high agreement for the statements" I compare prices of alternative winter destinations" (82%) and "I consider other winter destinations I could visit that fit into my travel budget" (83%). Compared to the loyal travellers from G1, the second group acts more independently, using less advice from friends "I compare posts about winter destinations with friends using Facebook, WhatsApp or similar" (34%) or from travel agencies "I visit travel agencies for advice about innovations/new winter destinations" (38%). Therefore, in G2 the focus is clearly on destination price and performance. In general, G3 shows a significantly lower interest in new winter travel products, collecting relevant information on the Internet or visiting a travel agency. Regarding prices, both undecided and disloyal travellers have a higher price sensitivity than loyal travellers.

The final section of the quantitative survey addressed two climate change questions: the observation of impacts from climate change and the information sources used for information on climate change in the Alps. As the impacts of climate change during winter in the Alps depend on altitude and region and therefore vary greatly among the destinations, this section first asked about the Alps as a whole and second about the preferred Alpine winter destination. All the questions were asked using a binary scale (yes/no) which allowed us to calculate the agreement within each group as a percentage and to compare groups using the two-sample *t*-test for independent variables. Table 7 shows the results in detail.

The questionnaire section about climate change started with two questions about the weather. The terms climate and climate change were not used in order to avoid socially desirable responses. The first question was: "Please remember the last 10–15 years and tell us: has the winter weather in the Alps changed significantly during that period?". If the answer was yes, the next question was: "Has the winter weather in your preferred Alpine destination changed during that period?". The first question was answered with yes by about two thirds of the interviewees (68%). Within this group, 88% reported that they had observed the change in winter weather in their preferred destination. This suggests that the majority of participants based their judgment on personal observation made during winter holidays in the Alps. Compared to the other two groups, the loyal group G1 reports a significantly higher number of periods with extreme snowfall—both for the Alps in general and in the preferred winter destination. Undecided travellers (G3) observed less snow than in the past for both the Alps and the preferred destination. Disloyal guests (G2) reported less often than G1 and G3 that the first snowfall in the Alps was arriving later and later in the season. Travellers usually see their personal observations confirmed by information they get from further sources: levels range between 73% (G2) and 96% (G3). All three groups (G1, G2 and G3) appear to use widely the same sources of general information on the climate in the Alps, even though at slightly different levels of intensity.

**Table 7.** Climate change: observation and information sources.

| | Agreement in % | | | G1–G2 | | G1–G3 | |
|---|---|---|---|---|---|---|---|
| | **G1** | **G2** | **G3** | *t*-Value | Sig. [a] | *t*-Value | Sig. [a] |
| Please remember the last 10–15 years. Please tell us: has the winter weather in the Alps changed? significant during that period? | 63.9 | 70.2 | 71.6 | | | | |
| *only cases with agreement to above statement: which observations did you make (Alps in general)* | | | | | | | |
| less snow—Alps in general | 87.5 | 88.0 | 94.2 | | | −1.744 | 0.041 ** |
| first snowfall comes later—Alps in general | 88.8 | 83.3 | 89.1 | 1.755 | 0.040 ** | | |
| snow melting starts earlier—Alps in general | 61.2 | 62.0 | 60.1 | | | | |
| periods with extreme snowfall—Alps in general | 47.4 | 40.0 | 35.5 | 1.437 | 0.076 * | 2.103 | 0.018 ** |
| periods with rain instead of snow—Alps in general | 76.3 | 72.7 | 75.4 | | | | |
| *how did you get aware about climate change in the Alps?* | | | | | | | |
| from news/newspapers/other media | 69.1 | 63.6 | 62.3 | | | 1.293 | 0.099 * |
| by the internet/social media | 41.4 | 45.7 | 39.9 | | | | |
| by reports of friends and relatives | 65.1 | 66.2 | 61.6 | | | | |
| own observations during trips to the Alps | 87.5 | 87.4 | 88.4 | | | | |
| own observations in my winter destinations | 82.9 | 72.8 | 87.7 | 1.613 | 0.054 * | | |
| Has the winter weather in your preferred Alpine destination changed during that period? | 89.5 | 83.0 | 91.4 | | | | |
| *only cases with agreement to above statement: which observations did you make (preferred destination)* | | | | | | | |
| less snow—my preferred Alpine destination | 87.5 | 92.0 | 95.2 | | | −1.716 | 0.044 ** |
| first snowfall comes later—my preferred Alpine destination | 86.8 | 84.8 | 87.2 | | | | |
| snow melting starts earlier—my preferred Alpine destination | 58.8 | 58.4 | 59.2 | | | | |
| periods with extreme snowfall—my preferred Alpine destination | 49.3 | 41.6 | 34.4 | 1.354 | 0.088 * | 2.553 | 0.006 ** |
| periods with rain instead of snow—my preferred Alpine destination | 76.5 | 73.6 | 72.0 | | | | |
| *how did you find out about climate change in your preferred Alpine destination?* | | | | | | | |
| from news/newspapers/other media | 67.6 | 59.8 | 62.7 | 1.316 | 0.095 * | | |
| on the internet/social media | 40.4 | 44.9 | 38.9 | | | | |
| through reports of friends and relatives | 65.4 | 66.9 | 61.1 | | | | |
| own observations during trips to the Alps | 89.0 | 89.0 | 90.5 | | | | |
| own observations in my preferred winter destination | 92.6 | 86.6 | 96.0 | 1.613 | 0.054 * | | |

[a] significance calculated as 1-tailed *t*-test; ** if smaller the 0.05, * if smaller than 0.1.

## 4. Discussion

This study investigated the role of climate change as one among several factors influencing destination choice and therefore destination loyalty. With 14.6 million skiers, Germany is the largest market followed by France (8.6 million) and UK (6.3 million) [50] in Europe. This is why Alpine winter vacationists from Germany were chosen for this study. Many studies have already discussed the impact of climate change on Alpine winter destination supply (Steiger and Mayer [6], Uhlmann et al. [4], Berghammer and Schmude [51]) as well as on demand (Steiger [52], Landauer et al. [53], Steiger [54]). Many of these studies assume that climate change is one single factor influencing destination choice and loyalty. Others are based on a limited set of potential factors because they use discrete choice modelling (Luthe and Schläpfer [55]), a method which does not allow a more holistic approach. Our research widens the perspective by trying to identify and consider each type of factor influencing loyalty. Research on destination loyalty mainly focuses on explaining what causes repeat visits. Several studies show that guest satisfaction plays a central role (Tasci and Gartner [56], Chi and Qu [57], Tasci [58]). In our study, we turn the perspective around and analyse reasons for terminating loyalty. Dissatisfaction is clearly a trigger, so one central question is the extent to which climate change contributes to dissatisfaction.

First, a qualitative study identified general elements which potentially have an impact on the decision for a change of destination. This allowed us to look for connections between winter vacations and the impacts of climate change. Second, a quantitative survey based on the results from the qualitative study analysed climate change as one among many other triggers leading guests to turn their backs on their preferred winter destination. Winter sports, especially Alpine skiing, are a regular part of many guests' activity patterns. By considering all types of activity patterns—and not only skiing—, our approach incorporates the latest research results of Crouch et al. [34], a correlation between destination image, past experiences and activity profile. As climate change might negatively impact expected activities, our study also covers the influence of climate change on the satisfaction of different types of guests.

The results from the qualitative study show that there are many potential factors causing a decline in loyalty. They confirm the importance of irregular incidents in a traveller's daily life such as the birth of a child, a change in partnership, a new job or the end of an illness. These are elements beyond the reach of the influence of destination management, and they are obviously independent of climate change. However, in this context, we also found negative attitudes regarding tourism's potential to affect the environment, specifically Alpine winter sports. These negative views may have increased since autumn 2018, as a result of the new social movements such as "Fridays for Future".

Looking back at the last 15 years, many study participants highlight landscape changes caused by skiing infrastructure. They say they feel uncomfortable with the omnipresent ski lifts and the high density of snowmaking facilities along the slopes. Furthermore, we found that some travellers associate negative experiences caused by rainy weather or lack of snow with climate change in the Alps. At the qualitative level these results confirm that climate change might be a trigger for leaving the Alps after a long period of destination loyalty. However, not everyone criticizes large-scale snowmaking in ski resorts. Ski enthusiasts accept this as a necessary adaptation measure in the face of climate change to maintain the basis for their preferred winter holiday activities. Furthermore, snowmaking offers a stable slope quality and makes the start of the season more reliable and predictable. Looking back, many remember earlier years when natural lack of snow made skiing impossible even in high altitude locations. They therefore appreciate the improvement, but they also see the negative effects, especially the progressive rise in ticket prices. These findings correspond with the results of the Swiss study conducted by Pütz et al. [41].

Climate change plays a double role when skiing is high on a tourist's activity profile (see Crouch et al. [34]). Real ski enthusiasts appreciate the guarantee of a great skiing experience, while guests with a more multi-optional profile weight the increasing cost against limited benefits. While investments in snow security might pay off to maintain the horizontal loyalty of skiing guests in Alpine

destinations, price-hikes weaken the destination relationship for other guest groups. In the long term this is problematic, as demographic change will reduce the overall numbers of skiers: the birth rate is decreasing and nowadays fewer children learn to ski. In addition, the qualitative study confirmed that the travellers who are starting to doubt in Alpine winter holidays often realize that other destinations offer many attractive alternatives, often at reasonable prices.

The results from the quantitative study along the model used in Figure 1 give us a more detailed understanding of the traveller variables as described by Woodside and Lysonski [42]. We first analysed the socio-demographic variables such as age, income or number of children but did not find statistically significant differences between the loyal and the two leaving groups. However, we were able to confirm that most of the irregular incidents recorded in the qualitative study were triggering events. This makes it clear that it is not the personal situation itself (e.g., the phase in the life cycle, the family situation or the profession) that leads to travellers reflecting on and changing their destination choice set (Decrop [32]). However, one or several of these aspects can be catalysts. Furthermore, we found that the impact of these catalysts was much stronger for the group "undecided" than for the group of "disloyal" travellers. This allows the hypothesis that undecided travellers often still have a high preference for Alpine destinations, but constraints resulting from major changes in their lives prevent them from returning. Good relationship management with this group could remind them of their positive experiences in the past and bring them back to the Alpine destination once the personal constraints have been overcome. These triggers are independent of climate change and will always exist.

Winter guests have high expectations of their Alpine holiday destinations. In total, 21 out of 48 destination attributes for typical Alpine winter destination experiences were rated 'highly important' by over 75% of the study participants. All the other attributes—except après-ski and ice-skating—were rated 'very important' by at least half of them.

The attractivity of Alpine destinations lies in their natural environment, which is a public good. The outstanding qualities and the uniqueness of this public good compared to other European winter destinations justifies the relatively high prices. This result is in line with the hedonic pricing theory described by Rigall-I-Torrent and Fluvià [59]. Some studies have used hedonic pricing (Falk [60] or Rosson and Zirulia [61]) to set up models for lift ticket prices in the Alps. These models relate lift ticket sales and prices to qualities of the Alpine destination. However, they only consider the qualities that are directly related to winter sports (length and altitude difference of ski lifts, length and altitude difference of ski runs, etc.). They do not take the surrounding scenery or the quality of the overall environment into consideration.

Our study looks deeper into guest expectations. It clearly shows that the overall environment plays a significant role above and beyond the skiing facilities. The ski lift operators can invest in creating more reliable snow conditions, faster and larger lifts and ever more exciting attractions on the mountainsides. However, the destruction of the scenery and the natural environment does not go unnoticed. As the models do not take this into account, they calculate that customers will be willing to pay higher prices for the benefits of higher investments in technical innovations. Our study questions this conclusion, because it shows that the customers who are turning away from Alpine destinations do not accept deterioration of nature in combination with rising prices. The PCA based on the fulfilment of traveller's expectations identifies exactly these elements as independent factors: basic destination quality and winter sports infrastructure, i.e., private goods, and pure nature and pristine Alpine landscape, i.e., public goods. Therefore, hedonic pricing models should be refined and integrate nature and landscape quality as independent variables to a much higher extent. Studies (see Hernandez et al. [62]) have shown that the surrounding environment of any component of a tourism destination is of significant importance to its value. Therefore, hedonic pricing using spatial econometric models or geographically weighted regression as proposed by Hernández et al. [62] might be more adequate for Alpine winter destinations.

We also looked for features of climate change that leave expectations unfulfilled and thereby create dissatisfaction leading to a change in destination loyalty. We found that features directly related to climate change led to significant levels of dissatisfaction, especially in the group of "undecided" travellers (snow security, experiencing snowfall, snow-covered landscape or frozen lakes). A further source for dissatisfaction were overcrowded slopes and the impression that skiers did not show enough respect for other people on them. Furthermore, during warm weather periods skiers switch to the rare slopes at higher altitude to find good snow conditions. This leads to even higher crowding effects.

A very high share of travellers was dissatisfied with the prices for ski lift tickets and equipment rental during their last stay. They paid a high price and expected perfect skiing conditions. This confirms the results of Fonner and Berrens [63], who showed that price acceptance decreases when individually perceived crowding exceeds thresholds. A new aspect in our research is that the perception of crowding is not only a question of demography, first-time versus repeat visitors, or crowding prevention strategies as described by Zehrer and Raich [64]. Zehrer and Raich found that first-time visitors perceived crowding to a higher extent than repeat visitors. Our sample, based only on repeat visitors to Alpine winter destinations, shows that the general level of satisfaction correlates with crowding perceptions. Dissatisfied guests leaving a destination tend to perceive crowding more frequently than satisfied and therefore loyal guests. This allows the hypothesis that the expectation level of guests in pure nature, unspoiled pristine landscape and quietness also have an influence on crowding perception. The results suggest that guests who are not primarily skiers perceive slopes more often crowded as skiing enthusiasts.

## 5. Conclusions

Taking a final comprehensive look at our results, the research question "Does climate change influence guest loyalty to Alpine winter destinations?" must be affirmed. However, climate change is not the only trigger turning loyal Alpine vacationists into disloyal or undecided travellers. For many guests in skiing destinations, the negative effects of adaptation measures outweigh the benefits in two ways: they contribute to the deterioration of nature and landscape aesthetics and they induce higher prices without a real added value. Therefore, Alpine destinations must start to think about exit strategies from an unlimited spiral of quantitative growth, and to set up zoning concepts and visitor management to re-create areas of peace and quiet and winter nature experience for non-skiers.

## 6. Limitations of Results

This study is based on a survey of German travellers to the Alps in the winter season. Consumer perceptions might differ in other European source markets for winter sports in the Alps—such as Great Britain, the Netherlands, Italy, France or Poland. The results might also be different for other major winter sports destinations such as the Carpathians or the Rocky Mountains. The PCA used in the survey explained 54.9% and 56.1% of the total variance using latent root criterion (eigenvalues > 1.0). The share of excluded and not interpreted variance therefore remains relatively high.

**Author Contributions:** Conceptualization, T.B. and A.H.; methodology, T.B. and A.H.; software, T.B.; validation, T.B., A.H. and S.G.; formal analysis, T.B., A.H. and S.G.; investigation, T.B. and A.H.; writing—original draft preparation, T.B., A.H.; writing—review and editing, S.G.; visualization, T.B.; project administration, T.B.; funding acquisition, T.B.

**Funding:** This research was funded by the Federal Ministry of the Environment, Nature Conservation and Nuclear Safety FKZ UM17163150.

**Acknowledgments:** The authors thank Patricia East from Munich University of Applied Sciences for checking the manuscript and contributing to the improvement of the paper by raising substantial questions and making suggestions to increase strictness of argumentation.

**Conflicts of Interest:** The authors declare no conflict of interest.

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
