# Peer review of "Does Climate Change Influence Guest Loyalty at Alpine Winter Destinations?"

_sustainability, doi:10.3390/su11154233_

Round 1
Reviewer 1 Report
Thanks very much authors for this interesting piece of work. I would like to recommend a few alterations to enhance the quality of the manuscript.
I am okay with the quality of the work. However, I would like to see the results of the statistical analyses through a few graphs and corresponding brief write up.
Author Response
Dear reviewer,
thanks a lot for your helpful remarks and suggestions we appreciated very much. We went point by point through them. Please find in the attached review report how we integrated them in the revised version.
We hope that our revisions and remarks find your approval.
Kind regards
Thomas Bausch
Andreas Humpe
Stefan Gössling

Reviewer 2 Report
The authors present an interesting study linking climate change and tourism. In my opinion, the paper is sound scientifically and I only made minor observations, here under.
on line 40-41, "studies has also ..." shouldn't it be "studies have also ..."
I note that the question "how does climate change influence destination image" not answered well
on line 114 "Half the participants ..." shouldn't have been "half of the participants ..."
I note that the study was conducted in Germany but the overview of the climate of Germany is not presented.
It also invites a question: how has the climate of Germany changed?
line 172-175 seems to show that the authors have limited distinction of the terms weather and climate. The authors should know that climate change is not necessarily change in weather
Table 2 fails to show the link of weather as a driver of change
on lines 185-188, I'm not comfortable with the ambiguity in using the term "several" when reporting results
I find lines 189-202 very subjective.
Author Response

(The authors gave the same response as above.)

Reviewer 3 Report
Dear author/s,
your paper is interesting, but before being considered for publishing I have a few recommendations:
please add a literature review section, by this way the gap in the literature will be better emphasize
please separate the results from the methodology (the steps of the research should be in the methodology part, the structure of the questionnaire etc.)
Please mention how you test the reliability and consistency of your model?
Please report the main results of the PCA (KMO, total variance etc.)
Pay attention to table 6, to the format of the numbers if the data are reported by percentage.
In the conclusions sections I recommend you to emphasize the theoretical implications of your study, managerial aspects and future research directions.
Good luck!
Author Response

(The authors gave the same response as above.)

Round 2
Reviewer 3 Report
Dear author/s,
thank you for the improved version of the manuscript. I still have one concern regarding the level of the total explained variance, which is below the recommend level of 60% (Hair et al. 2014), since you used a model well known and tested. Please mention it as a limitation of your study.
On the other hand, please pay attention to the format of the paper.
Good luck!
Author Response
Thanks for the suggestions concerning a more clear expressing the limitations of our PCA. We studied the reference (Hair et. al. 2014) and the section you referred at:
PERCENTAGE OF VARIANCE CRITERION The percentage of variance criterion is an approach based on achieving a specified cumulative percentage of total variance extracted by successive fac- tors. The purpose is to ensure practical significance for the derived factors by ensuring that they explain at least a specified amount of variance. No absolute threshold has been adopted for all applications. However, in the natural sciences the factoring procedure usually should not be stopped until the extracted factors account for at least 95 percent of the variance or until the last factor accounts for only a small portion (less than 5%). In contrast, in the social sciences, where information is often less precise, it is not uncommon to consider a solution that accounts for 60 percent of the total variance (and in some instances even less) as satisfactory.
Having 54.9% and 56.1% we are below the level mentioned by Hair et. al. as a criterion for a satisfactory solution. The authors know that there is still plenty of information in the data which is not yet part of the interpretation as it is excluded by the PCA. We do not see this as a general deficit, as it also shows that the used items were well chosen and many of them measure independent aspects. As the paper already now has overlength we resigned from discussing factors with eigenvalues below but near to 1.
Nevertheless we agree, that a explicit note about the limitations of our PCA is useful for the readers. Therefore we added a section 6. Limitations of Results including this aspect.
Thanks again for the reviewing and your contributions to improve our paper.
Round 3
Reviewer 3 Report
Dear author/s,
Thank you for your answer.
Good luck!